# High Density Real-Time Air Quality Derived Services from IoT Networks

**DOI:** 10.3390/s20185435

**Published:** 2020-09-22

**Authors:** Claudio Badii, Stefano Bilotta, Daniele Cenni, Angelo Difino, Paolo Nesi, Irene Paoli, Michela Paolucci

**Affiliations:** Department of Information Engineering, DISIT Lab, University of Florence, 50139 Florence, Italy; Claudio.badii@unifi.it (C.B.); stefano.bilotta@unifi.it (S.B.); Daniele.Cenni@unifi.it (D.C.); angelo.difino@unifi.it (A.D.); irene.paoli@unifi.it (I.P.); michela.paolucci@unifi.it (M.P.)

**Keywords:** smart city, pollution, interpolation, IoT (Internet of Thing) application, dashboards, sensors network, early warning system, devices dysfunction

## Abstract

In recent years, there is an increasing attention on air quality derived services for the final users. A dense grid of measures is needed to implement services such as conditional routing, alerting on data values for personal usage, data heatmaps for Dashboards in control room for the operators, and for web and mobile applications for the city users. Therefore, the challenge consists of providing high density data and services starting from scattered data and regardless of the number of sensors and their position to a large number of users. To this aim, this paper is focused on providing an integrated solution addressing at the same time multiple aspects: To create and optimize algorithms for data interpolation (creating regular data from scattered), making it possible to cope with the scalability and providing support for on demand services to provide air quality data in any point of the city with dense data. To this end, the accuracy of different interpolation algorithms has been evaluated comparing the results with respect to real values. In addition, the trends of heatmaps interpolation errors have been exploited to detected devices’ dysfunctions. Such anomalies may often be useful to request a maintenance action. The solution proposed has been integrated as a Micro Services providing data analytics in a data flow real time process based on Node.JS Node-RED, called in the paper IoT Applications. The specific case presented in this paper refers to the data and the solution of Snap4City for Helsinki. Snap4City, which has been developed as a part of Select4Cities PCP of the European Commission, and it is presently used in a number of cities and areas in Europe.

## 1. Introduction

The city users are increasingly aware of the health and economic costs of air pollution. Poor air quality is linked to over three million deaths each year; 96% of people in large cities are exposed to pollutant levels that are above recommended limits [1]. The costs of urban air pollution amount to 2% of gross domestic product in developed countries and 5% in developing countries [1]. For these reasons, most of the cities and regions are increasing their attention to the real-time monitoring of environmental and weather parameters [2,3]. In the past, the usage of environmental data sensors was in most cases a prerogative of public administrations. The main difficulties have been due to the objective criticisms in providing precise answers to the values of pollutants and aerosol since collecting values from a very limited number of sensors for a whole city does not mean that those values are valid for the whole city. In fact, it is reasonable to have large differences in some values when the air quality data is measured on a high traffic road rather than in a garden, for example, in the back of a house located on the same road. Recently, there is a deeper understanding of the environmental parameters (for example, $PM_{10}$ (Particulate Matter), $PM_{2.5}$, *CO* (Carbon Monoxide), $CO_{2}$ (Carbon dioxide), $SO_{2}$ (Sulfur dioxide), $O_{3}$ (Ozone), $H_{2}$*S* (Hydrogen Sulfide), *NO* (nitric oxide), $NO_{2}$ (Nitric dioxide), and $NO_{x}$ (Nitric monoxide and dioxide)), and how much they are influenced by the city’s structures, what the causes of high values of pollutants and aerosols are, the reasons for the registered high values, and the dynamic of their diffusion/propagation, etc. Thus, cities have a strong interest in understanding how much pollution affects the quality of the air that citizens breath, in order to properly regulate urban mobility, as well as other activities, and give city users awareness that they are living in a city that keeps them informed, is technologic oriented, and focused on citizens’ health and quality of life, as realised also in [4].

In the last few years, the great diffusion of low-cost sensors has given rise to the diffusion of many researches in the Smart City field. On one hand, to make citizens participate in the activities of their city and to make them aware of environmental issues and, on the other hand, to give citizens a series of low-cost sensors in order to have a multitude of data coming from sensors able to detect air quality, and to organize crowdfunding campaigns [5]. Obviously, the data coming from such a network of sensors must be validated and compared with those of the official stations, that often make available the measured data via some Open Data license. Even if the low-cost sensors are usually of low quality, the increased number of data, and the procedures for their calibration, can compensate for their shortcomings [6]. Citizens often welcome such types of campaigns with great enthusiasm because they are interested in having real time, local information, and maybe even forecasts: For example, to know the air quality at home and around their building, to know if the bike paths in the city centre are a good choice or if it is better to walk or to bike in parks; to know if their children go to school in more or less polluted areas of the city; and to choose a path for jogging [7].

The above described demands are diffused among the final users, which request to have access to air quality status with general views as heatmaps and on specific points and paths, where they live, walk, or run. The air quality parameters influence their decisions on routing, walking, and in selecting locations to spend time with family or to participate in sport. Therefore, they are also of fundamental importance in the definition of the urban strategy plan on which city decision-makers have to work, for cutting costs and time, exploiting advanced tools for control room and business intelligence as dashboards [8]. For these reasons, this paper presents a scalable solution for creating smart services exploiting sensors’ data regarding air quality parameters and providing heatmaps and punctual information to a large number of city users that typically request them via mobile Apps.

### 1.1. Related Work

Nowadays, the estimation of the concentration of air pollution in smart cities is an important issue. Traditional methods for air quality monitoring suffers from limited spatial coverage and granularity due to the high costs of construction, installation, and maintenance and thus can be only arranged on a limited number of points. A dense air quality information map can be obtained from data collected by mobile sensing systems, by which measurements could also be taken inside buildings (e.g., parking silos) thus producing evident problems. However, these methods still cannot achieve high accuracy because the number of devices is still too small to cover the whole area and the time schedule of mobile vehicles leads to coverage change over the course of the day [9].

In order to monitor air pollution starting from scattered data, it is important to estimate pollution levels at any time and location. The literature dealing with the monitoring of air quality and health condition is extensive and related to several contexts. In particular, the literature contributions can be traced back to in two macro-areas. The first area is related to air pollution forecasting and early warning systems [1,10,11,12,13]. For example, in [12] the authors adopted the cloud computing platform Azure, which has the advantages of scalability, reliability, and agility, to set up a multi-pollutant air quality deterioration warning system. They collected and integrated the government and private data into the database with the aim to predict multipollutant air quality. The second area is related to real-time pollution monitoring through interpolation methods and visualization systems. The use of deterministic and stochastic interpolation methods to estimate unknown values such as Inverse Distance Weighting (IDW), Kriging [14], spline, radial basis function, natural neighbor or trend surface are widely used in literature. Only a few studies explored a system for a real-time fast air pollutant visualization [3,15], especially at a large geographic scale. In [9], the authors collected data from 11 mobile sensing units on vehicles in Tianjin for 1 month to cover the entire area; thus, their system suffered from problems such as lack of data and high uncertainty because the data amount and distribution vary over time. To address these problems the authors combined two classic data driven models, Kriging and IDW. They adopted the Random Forest Algorithm to adaptively choose the more accurate models (Kriging or IDW). In [16] the authors evaluated two interpolation methodologies: Artificial Neural Networks and Multiple Linear Regression, using data from a real urban air quality monitoring network located at the greater area of metropolitan Athens in Greece. The statistical analysis was based on the development of an air quality database of hourly air pollutant concentrations for examining the intra-daily spatial variability and the variations in average annual pollution values but not a real time evaluation and visualization of the situation at all points in the area. In [17], the authors compared Shape Function (SF)-based and IDW-based spatiotemporal interpolation methods on a data set of PM2.5 data in the contiguous United States Particle pollution. They applied a cross validation technique to validate the interpolation and consider historical data (146,125 daily measurements at 955 monitoring sites in 2009) without providing a real-time service. A mapping of the distribution of pollutants was estimated in [18], in order to reveal the relationship between them and the demography of the region. In [19], the authors developed a web application for depiction of air quality using heatmaps by exploiting a cubic spline interpolation method. Their production of heatmap uses the Application Programming Interfaces (APIs) provided by Google Maps.

A further aspect that could undoubtedly interest both citizens and public administrations, is to connect the level of air quality with the type of district of the smart city. Each district is often dominated by commercial activities, green spaces, long/medium distance streets, industrial areas, etc. In [20], a study was carried out on AQI (Air Quality Index) considering the various pollutants and bearing in mind the points of interest in a Smart City, where each sensor was positioned, although in reality the research was based on the quality of AQI prediction from the type of sensors used and no real study was made to correlate the type of pollutant that most affects the calculation of AQI and type of area of the city (e.g., commercial, residential, green, city center, RTZ (Restricted Traffic Zone). In [21], a comparison among the AQI and the different city contexts was estimated, and the study was limited to roadside and city background situations. The study realized in [22] defined a smart city divided in urban functional zones (agricultural, industrial, commercial, multi dwelling housing, individual housing) which were described and categorized in detail, though a correlation among each of them and the specific pollutants emitted and air quality estimation was missing. In [23], a study on AQI estimation was realized connected to the city morphological characteristics of a smart city, the study was mainly focused on density population, road graph, buildings, Point of Interest (POI), traffic data, vegetation, and meteorological data. However, the research, even if it took into account many features of different types that outline a smart city, was carried out only on a statistical basis and was therefore not accompanied by a validation of data and forecasts obtained in real time.

### 1.2. Paper Contributions

This paper presents four main contributions. The first contribution consists in presenting an effective solution for creating dense heatmaps from scattered data and for providing smart services on dense grid such as conditional routing on pollutant, alerting/warning on the basis of pollutant values, providing users with general information via mobile app and dashboards, may be as a support for what-if Analysis. In this context, this paper reports different algorithms and a scalable solution for computing pollutant-related smart services.

The second contribution regards the comparison of different algorithms for computing dense grid values of pollutants, based on the scattered sensor data coming from the IoT (Internet of Thing) sensors. The assessment of the estimation error for the dense grid of values, starting from scattered points of the sensors, gives a measure of the resulting data quality. In certain cases, the pollutant values are strongly influenced by the city’s structure, by the wind, etc. The error analysis allowed us to validate the solution.

The third contribution is related to the second one. Since the error analysis allowed us to validate the solution, we exploited the error analysis as a tool for detecting device failures. On the basis of the error analysis it was possible to understand anomalies on devices, which helped to avoid taking into account sensors’ data that could compromise the service quality. In the cases of failure detection, a warning was sent to a tool, called Data Ingestion Process Manager, and to the maintenance team.

The fourth and last contribution consists of providing a solution for simplifying the production process of integrated data analytics in data flow real time processes called, in this paper, IoT Applications. The solution and tools permitted to develop the other results and thus for developers to create their own data analytics algorithms and see them exploited as a basis of smart solutions (e.g., for computing heatmaps and predictions). This means that the data analysts can focus on the algorithm while the integration with Smart City APIs, scheduling, and management of the whole workflow are automatically managed by the solution that can be easily applied.

The above results have been developed and integrated into the Snap4City solution and infrastructure. Snap4City is 100% open source and allows to ingest and manage Big Data coming from IoT devices, applications, and services; compute actions for users, providing notifications and engagement; and produce interactive dashboards supporting decision-making processes (useful for many different kinds of users: Public administrators, decision makers, final users, city operators, etc.). Ingested data are collected, aggregated, and indexed in the Snap4City Knowledge Base, which is based on the Km4City multi domain ontology in order to speed up and facilitate data retrieval actions [15,24]. Snap4City supports the creation of data-driven applications, based on MicroServices in Node-RED (https://nodered.org). The Snap4City solution can also be used for developing mobile and web apps, data flows, and data processing tools [25].

### 1.3. Paper Structure

The paper is organized as follows. Section 2 describes the Snap4City Framework and Development facilities. In detail, Section 3 reports the detailed data and process flows for the smart air quality services developed, and the details regarding the creation and optimization of algorithms for data interpolation for heatmap production. The solution has been generalized to be used on a large range of sensor data of environmental variables such as $PM_{10}$, $PM_{2.5}$, *CO*, $CO_{2}$, $SO_{2}$, $O_{3}$, $H_{2}$*S*, *NO*, $NO_{2}$, $NO_{x}$, air temperature, air humidity, velocity of wind speed, and dew point. The smart services are computing green path routing (for jogging, walking with kids or with highly sensitive people, etc.), sending personal warnings on specific points, representing heatmaps on mobile and web page, etc. Section 4 provides a description of Helsinki’s city use case scenario, adopted to identify and validate the models and the framework. It is important to note that in Helsinki the official sensors’ network for air quality assessment has been enriched with data coming from IoT devices installed and manage by city users. The resulting data and heatmap area was made accessible to the final users via dashboards and mobile applications. In Section 4.1, the details regarding the IoT devices and network are reported. In Section 4.2, the different interpolation algorithms are compared, and the errors of interpolation assessed. In Section 5, the approach used for detecting anomalies that can be useful for discovering any problems related to device sensors, and thus for increasing the quality in computing smart service data, is described. The automatization of the heatmaps’ creation/customization is presented in Section 6. This automatic process simplifies the production of integrated data analytics in data flow real time processes, called IoT Applications. Conclusions are drawn in Section 7.

## 2. Snap4City Framework and Development

The solution proposed in this paper has been developed by exploiting Snap4City architecture (see Figure 1), and its main elements are described below, addressing data ingestion and data analytics aspects that are functional to the explanation of the solution presented in this paper. The Data Ingestion and Aggregation can be performed by using: IoT Applications in Node-RED, while other tools as CKAN for open data (https://ckan.org), and Web Scraping tools. The solution supports a large number of protocols [26] such as MQTT (Message Queuing Telemetry Transport), NGSI (Next Generation Service Interfaces), COAP (Constrained Application Protocol), OneM2M (Machine to Machine), ModBus (Modicon BUS), OPC (Open Communication), and AMQP (Advanced Message Queuing Protocol). Data Transformation tools for manipulating data are mainly developed by using data flow, data driven flows in Node-RED exploiting a suite of Snap4City MicroServices. The Data Storage is implemented by using NIFI for data integration, Elastic Search for indexing, and the Knowledge Base implemented as an RDF (Resource Description Framework) store which is an index for spatial, relational, and temporal aspects. All the data and services are accessible via the Advanced Smart City API which are used by Front End Tools such as Dashboards, Web and Mobile Apps [26]. A Living Lab Development and Management layer provides on-line tools to developers and stakeholders to implement ingestion and data processing processes such as data flows, data analytics algorithms, Dashboards, IoT Applications, and Web and Mobile Apps [15]. In the architecture of Figure 1, the back-office tools have not been drawn, while Snap4City also presents a large set of tools for managing users’ activities, platform setup, auditing, assessment, control, monitoring, data registration, process management, security, scaling, and which are out of the context of this paper.

### 2.1. Data Transformations and Data Ingestion Processes

In Snap4City, a number of facilities allows to implement processes for data transformation and user interaction for Smart Services. The data arrive in Push or Pull and are collected by ingestion processes, that save and read data into/from the Big Data Storage. This approach automatically realizes the so-called Data Shadow or Data Hub in other solutions such as AWS (Amazon Web Service), Azure IoT, and Google Cloud IoT. With the aim of developing smart services, one must typically implement patterns in which the service on the consumer side need to:Access at contextualized data where the context includes GPS (Global Positioning System) position. To some extent, several kinds of data analytics can be computed in advance (i.e., periodically scheduled) and provided to all requesting users. Therefore, they are not computationally intensive (e.g., maps, heatmaps, POI, alerts) if their estimation is needed only a limited number of predefined area and times per day. For example, the status of parking is relevant in real time or prediction, and the same small piece of information is relevant for many people, while routing avoiding specific pollutants may lead to estimating them every few meters, for each path and for each person;receive notifications with some of the above-mentioned information—e.g., predictions, suggestions of pollutants. For example, the user may ask the system to inform them when a given pollutant is over a threshold or when the dew point is critical. This approach is a sort of service subscription and may be computationally intensive since the numbers of users may be high and the number of times in which the values have to be controlled can be high; andrequest specific on demand services such as routing, querying, what-if analysis [Pearl, 2019], and simulations. In most cases, these requests directly need to activate some data analytic to compute results as input to the decision making processes in the city management. Simulations could be related to the computation of emission in the new conditions, etc.

This means that a smart service development environment has to provide support for the creation of both scheduled and data driven processes, and that the connection from the final user front-end should be easily realized. From the point of view of data transformation developers, the processes are executed in a cluster of processing containers, in a form of IoT Applications (i.e., data flows’ formalization, periodic or data driven), can access and save data by using MicroServices [26]. In Snap4City, a collection of more than 150 Smart City MicroServices has been developed as Nodes for the Node-RED programming tool. Node-RED’s visual programming allows to develop IoT Applications exploiting both Push and Pull data.

### 2.2. Supporting Data Analytic Development

For the data analytic development, it has to be possible to access the Big Data store respecting the privacy and the data licensing by using authenticated Smart City APIs. The access has to permit to read historical and real time data, and to save the resulting data provided by the algorithms, for example, heatmap-related predictions, the assessment of data quality, and labels of detected anomalies. The work of the data scientist could finish once the algorithm has been developed which he/she should be aware of. On the other hand, the same algorithm (e.g., for computing heatmaps, parking prediction), should allow to be:Used on different services of the same kind located in different places and on the basis of a number of parameters (e.g., target precision and list of data sources). This means that data analytic itself has to be designed with the needed flexibility and generality;put in execution from IoT Applications by passing a set of parameters and collecting the results on the Data Storage or as a result of the invocation. The executions can be periodic or event driven—e.g., the arrival of a request or by the arrival of the new set of data values;controlled for collecting eventual errors and mistakes, in debug and at run time for logging. This may be for informing the developer and/or the administrator of eventual mistakes and problems by sending notifications; anddynamically allocated on cloud in one or multiple instances to plan a massive computation of the same data analytic process on several data sets and services at the same time.

Therefore, the solution developed for Snap4City satisfies all the above described requirements. Data analytic processes can be developed using R Studio or Python. In both cases, the code has to include a library for creating a REST Call, namely: Plumber for R Studio and Flask for Python. In this manner, each process presents a specific API, which is accessible from an IoT Application as a MicroService, that is, a node of the above-mentioned Node-RED visual programming tool for data flow.

Data scientists can develop and debug/test the data analytic processes on Snap4City cloud environment since it is the only way to access at the Smart City API with the needed permissions. The source code can be shared among developers with the tool “Resource Manager”, that also allows the developers to perform queries and retrieve source code made available by other developers. The described approach for creating data analytics processes has been used for developing and computing heatmaps and other algorithms described in this paper.

## 3. Dense Map Estimation and Services Provided

As described above, the city users need to get values within a few meters of their location, since in many cases they walk and run in the city or in rural areas. Most of the requests which are sent by users are on specific paths and areas, and the majority of the points computed for highly dense maps would be not useful since they are only sporadically requested, if at all, for example, for computing green path routings (for jogging, walking with kids or with highly sensitive people), sending personal warning on specific points or visualizing heatmaps on mobile and web pages. For the last case, the resolution can be much lower than those for the first two cases.

Therefore, there is the needs of providing a data value at any provided GPS coordinates of the city, for a large number of pollutants as well as for weather data such as $PM_{10}$, $PM_{2.5}$, *CO*, $CO_{2}$, $SO_{2}$, $O_{3}$, $H_{2}$*S*, *NO*, $NO_{2}$, $NO_{x}$, air temperature, air humidity, velocity of wind speed, and dew point. The values collected from the sensors are typically scattered in the area, and too often far from the GPS points in which the service is requested. Thus, a solution could be to estimate, since the beginning, a very dense grid of values. On the other hand, in a large city or geographic area, the computation of a dense grid would be computationally expensive and storage demanding. For these reasons, we decided to find a compromise by computing an interpolated grid of values at the resolution useful for generating heatmaps on mobile Apps and Dashboards, which are the typical maps requested by the majority of users. Furthermore, we decided to compute on demand with a second level interpolation the values at any GPS coordinate when a city user requests them (the caching of these computations is also viable). This workflow is described in Figure 2. In this manner, the service can be provided to several thousands of users. In fact, the GeoTIFF heatmaps are computed only once and distributed via a GeoServer through the standard Web Map Service (WMS) protocol. In the cases in which the data are measured several times a day, it could be possible to request an animation of the heatmap produced during the day.

The “On Demand Fine Interpolation” of Figure 2 is activated at each request to produce results in real time exploiting the Heatmap Data Server, on the basis of:GPS position for certain point of interest (POI) of the users, for computing an eventual alerting message if the user subscribes for the specific pollutant or weather data. The service is activated for each variable and each POI of each user. The POI could be selected by the user on the mobile, making a subscription or on the Map (dashboard or on mobile) by picking on the map itself.Conditional routing (e.g., from–to geographical points and routes) to get back possible paths, minimizing the pollutant or weather variables. The service is activated for each possible path and road segment of each routing request of each user.

The following subsections present the description of two different solutions for the Computation of the Data Values on Grid and the solution adopted for the On Demand Fine Interpolation. It should be noted that reference data for a given area may be affected by severe errors. The lack of data from a certain device could restrict the area in which the results and services may be provided. Relevant errors/noise/anomaly in some devices can produce severe errors in the resulting services. Thus, to avoid problem we need to remove the problematic data from the list of the scattered data to be used for the computing.

During our research, different techniques for the computation of the Data Values on Grid were taken into account. As a result, we decided to use deterministic methods instead of statistical ones, due to the small number of measures (i.e., sensor data locations) to interpolate. For this reason, geostatistical methods such as kriging [27] had to be discarded a priori. For example, in the case of kriging, the data have to be used to choose a variogram or covariance function, which turns out to be very difficult in the case of a small number of data locations, resulting in insufficient information. Among deterministic methods, the Bivariate Interpolation method (Akima) [28], and Inverse Distance Weighting method (IDW) [29], Radial Basis Functions (RBFs) [30]. Also, in this case, RBFs are used to produce smooth surfaces from a large number of data points. Therefore, Bivariate Interpolation method (Akima), and IDW have been considered as the most appropriate for the solution. The following two subsections provide an overview of the interpolation techniques finally considered: Bivariate Interpolation method (Akima), and IDW.

### 3.1. Bivariate Interpolation Method

The creation of heatmaps for particulate matter may be based on a gridded bivariate interpolation for irregular data [28]. The bivariate interpolation method consists of five steps:Triangulation (i.e., area partitioning in a number of triangles). For a unique partitioning, the x–y plane is divided into triangles by the following steps. 1.1: Determine the nearest pair of data points and draw a line segment between the points. 1.2: Find the nearest pair of data points among the remaining pairs and draw a line segment between these points if the line segment to be drawn does not cross any other line segment already drawn. Repeat 1.2 step for all possible pairs;selection of several data points that are closest to each data point (sensor) and are used for estimating the partial derivatives;organization of the resulting data with respect to triangle numbers;estimation of partial derivatives at each data point; andcomputation *of* the interpolation at each output point.

In more details, the z value of the function in the point of coordinates *x-y* in a triangle is interpolated by a bivariate fifth-degree polynomial in *x* and *y*:(1)$$z\left(x,y\right)=\overset{5}{\underset{j=0}{\sum }}\overset{5-j}{\underset{k=0}{\sum }}q_{jk}x^{j}y^{k}$$
where: The coefficients are determined by the given z values at the three vertexes of the triangle and the estimated values of partial derivatives (i.e., z, $z_{x}$, $z_{y}$, $z_{xx}$, $z_{xy}$, and $z_{yy}$) at the vertexes, together with the imposed condition that the partial derivative of z by the variable measured in the direction perpendicular to each side of the triangle, must be a polynomial of third degree three; at least, in the variable measured along the side [28].

### 3.2. Inverse Distance Weighting Method

The IDW is a deterministic mathematical method widely used in the geoscience field [29]. The IDW is based on the premise that predictions are a linear combination of data. Each interpolated value of a point is identified as the following equation:(2)$$z\left(x,y\right)=\sum_{i=1}^{n}w_{i}z_{i}$$
(3)$$w_{i}={\left(1/d_{i}\right)}^{p}/\sum_{k=1}^{n}{\left(1/d_{i}\right)}^{p}$$
where: z(x,y) is the interpolated value at the location (x,y); $z_{i}$ is the observed value; $d_{i}$ is the Euclidean distance between the point i and the interpolated point; and $w_{i}$ is the weight for the point each point ($x_{i},\;y_{i}$) and (x,y). The parameter p is the power value, i.e., the exponent that influences the weighting of $w_{i}$ on z, [31]. In general, the value at which p is affects the rate of weight reduction. If p = 0, there is no decrease with distance and the prediction will be the average of all the data values in the search neighborhood. As p increases, the weights for distant points decrease rapidly; and if p is very high, only the immediate surrounding points influence the prediction. In this research, and thus in the following experiments, we have tested different values of the parameters. After previewing the output and examining the cross-validation statistics, we setup the power value of p equal to 2. When p = 2, the method is known as the inverse distance squared weighted interpolation.

### 3.3. On Demand Fine Interpolation

This subsection describes the solution adopted for the on-demand fine interpolation of Figure 2. The solution is based on performing an IDW interpolation within each square of the interpolated grid using Vincenty’s distance [32] according to the steps reported in the following pseudocode (Algorithm 1).


**Algorithm 1**
1: **Input**: $\left(x_{k},y_{k}\right)$ = GPS position in the picked location k at time t
2: **for each** iteration i=1 to 15
3:   **compute**
$b_{\left(x_{k},y_{k}\right)}$: a circular area around the picked location k of radius equal to 2 km
4:    **get**
$\hat{z}\left(t\right)=\left({\hat{z}}_{1}\left(t\right),\;\cdots ,{\hat{z}}_{j}\left(t\right),\cdots ,{\hat{z}}_{n}\left(t\right)\right)$ vector of interpolated   values at time t inside $b_{\left(x_{k},y_{k}\right)}$
5:    **if**
$\hat{z}\left(t\right)$ = Ø, means no values in $b_{\left(x_{k},y_{k}\right)}$
**then**
6:       radius$\left(b_{\left(x_{k},y_{k}\right)}\right)$ = radius$\left(b_{\left(x_{k},y_{k}\right)}\right)*$2
7:    **continue**
8:       **for each** GPS position $\left(x_{{\hat{z}}_{j}},y_{{\hat{z}}_{j}}\right)$ of $\hat{z}\left(t\right)$ vector
9:          **compute**
$d_{\left(x_{k},y_{k}\right),\;\left(x_{{\hat{z}}_{j}},y_{{\hat{z}}_{j}}\right)}:$ Vicenty distance between $\left(x_{k},y_{k}\right)$ and $\left(x_{{\hat{z}}_{j}},y_{{\hat{z}}_{j}}\right)$
10:         **if**
$d_{\left(x_{k},y_{k}\right),\;\left(x_{{\hat{z}}_{j}},y_{{\hat{z}}_{j}}\right)}$< 0.0000000001 **then**
11:           ${\hat{z}}_{k}$(t) = ${\hat{z}}_{j}\left(t\right)$
12:         **else compute**
${\hat{z}}_{k}$(t): interpolated value at time t in the *k*-th picked locations using IDW
method
13:         **end if**
14:      **end for**
15:   **end if**
16: **end for**

## 4. City of Helsinki’s Scenario and Final Users Tools

In this section, it is shown how the above-mentioned solutions have been applied in the City of Helsinki’s scenario in the case of environmental monitoring and innovative solutions. The environmental monitoring use case has been primarily performed in a new smart district called Jätkäsaari, a peninsula and a quarter placed in the south of Helsinki city center. In addition to 20,000+ future inhabitants and workplaces for 6000 people, including various hotels and office facilities, Jätkäsaari also encompasses the main part of Helsinki’s passenger harbor. The large construction sites, the intensive and obstructed traffic, and the growing population, create environmental challenges in Jätkäsaari. Thus, there was the need for a platform that could integrate data from different sources and services, provide tools for data analytics, and enable the development of new innovative services, to easily and quickly get reliable information about the current state of the air quality and other environmental indicators, in different parts of Jätkäsaari. There are ongoing activities to measure the air quality near construction sites, performed by the Helsinki Region Environmental Services (HSY). The concentration of inhalable particles ($PM_{10}$) is being measured near the Kalasatama School and other sensitive targets, such as day care facilities, playgrounds, primary schools, senior citizens’ housing and services, and hospitals. Smaller inhalable particles (less than 10 µm in diameter, ($PM_{10}$) are not noticeably visible but they can cause health problems. The measurement results describe the air quality in a residential area with several major construction sites in the immediate vicinity. Respiratory particulate concentrations are high when the daily average exceeds the limit value of 50 µg/m^3^. The air quality is poor when the hourly rate is above 100 µg/m^3^. In addition, a preliminary analysis has been conducted in order to understand the levels of each pollutant and the European Air Quality Index (EAQI), comparing the residential area of Jätkäsaari peninsula described above, with the Helsinki station area (downtown area). Table 1 reports the annual average values of $PM_{10}$, $PM_{2.5}$, $NO_{x}$, and EAQI.

Figure 3 reports the $PM_{10}$ average trend per hour for the period from September to December 2019 in Jätkäsaari area, considering 22 different devices (see for details next subsection). In this case, for the working days the average value is equal to 6.26, the median is 4.29 and the standard deviation is 9.68. For the weekend, the average value is 7.41, the median is 3.06 and the standard deviation is 16.6.

Therefore, in the context of Snap4City a number of Dashboards have been developed for the City Operators and ICT Officials, while a Mobile App has been developed and published both on the Google Play Store and the Apple Store for the Citizens and Tourists. The main Dashboard is reported in Figure 4a, while a detail regarding one heatmap is reported in Figure 4b.

The dashboard showed in Figure 4a can be found at https://www.snap4city.org/dashboardSmartCity/view/index.php?iddasboard=MTQwNg==. Moreover, the Mobile Apps allow the city users to access the heatmaps, and to subscribe to notifications that are activated on their POI when the selected pollutant is above the critical values.

Figure 5a,b report the two different visualizations of the heatmap presented in Section 3. In Figure 5a the Inverse Distance Weighting Method (IDW) was used to interpolate all the Helsinki area considering also those points on the map that are not included in the internal perimeter traced from sensors. The result is a regular map of the entire area of interest. In Figure 5b the bivariate interpolation method, Akima, has been applied for Helsinki and the result is an irregular map. The interpolated points are those inside the perimeter drawn by the sensors. Note that, some sensors are not included in the colored area because they do not produce a measure.

In Figure 6**,** two snapshots of the Mobile App “Helsinki in a Snap” are reported.

### 4.1. Sensors Description, Data Calibration, and Data Gathering

In Jätkäsaari peninsula, about 25 fixed devices have been installed and taken into account for the interpolation. They have been 25 AQ Burk low-cost (https://vekotinverstas.fi/aqburk) sensors realized with low-cost microcontrollers (e.g., ESP8266, ESP32) connected with particulate sensors (SDS011 laser-scattering PM sensor) and gas sensor (BME680 sensor). Typically, they have been installed in private buildings by Forum Virium Helsinki during a campaign to involve citizens in monitoring the air quality level in presence of a set of new building construction works. AQ Burk sensors measure PM2.5 and PM10 values once per second and their measurements are considered fairly reliable. The lifetime of SDS011 is max one year (8000 h), so it is expected that their measures would deteriorate and eventually stop sending data. The SDS011 particle sensor has a small fan that allows it to suck air. Inside the sensor, the light from the laser diode scatters air particles of different sizes into two detectors, from which the device calculates the PM10 and PM2.5 values of the air sample once a second. The device sends the data over the serial bus to the microcontroller for further processing.

The calibration has been performed both comparing the sensors values each other’s and comparing their values with the Helsinki Region Environmental Services Authority (HSY, https://gnf.fi/en/gnf/hsy_en) air quality measuring stations (moving the low-cost sensors near the HSY stations). After the calibration phase, it has been detected that sensors were sensitive to measurement errors due to humidity because they did not have sample air treatment (sample air drying or heating). During the period in which the measurements and research described in this paper have been realized, the sensors can be considered as fixed sensors, that is located in the same place.

AQ Burk sends data using LoRaWAN network to an IoT Broker, which stores the data locally into InfluxDB database and also forwards them via a NGSI API (at IoT Orion Context Broker at ngsi.fvh.fi) and to the Snap4City Platform IoT Broker (called orionFinland, in Figure 7a). When data arrive to the Snap4City Broker, they are divided in static data and real time data. The static data are registered in the Snap4City Knowledge Base and are visible both in Figure 7a,b. Figure 7a shows IoT Broker URI (of the Data Provider, e.g., https://ngsi.fvh.fi), Device Type according to KM4City multi-ontology (e.g., AirQualityObserved) and Type (e.g., sensor), Protocol used for receiving data (e.g., ngsi), GPS Coordinates (latitude and longitude), Device Uri on Snap4City Platform (e.g., http://www.disit.org/km4city/resource/iot/orionFinland/Helsinki/373773207E33012B), date of creation of the IoT Device in Snap4City, format of file received form Provider (containing the real time data), Producer (or Data Provider). In Figure 7b, the types of real time data for each sensor and the related unit of measures, frequency, etc., are listed (e.g., $PM_{10},PM_{2.5},$ reliability, and date observed for each measure). The Real Time data are gathered in Snap4City with a frequency of 1 min, and automatically associated with each IoT Device (AQ Burk sensors on Snap4City), as can be viewed in the data history trend shown in Figure 7c).

### 4.2. Validation in the Context of Helsinki Data

Each device described in Section 4.1 performs two different measures of particulate matter: $PM_{10}$, $PM_{2.5}$. For each kind of pollutant, the interpolated heatmap is computed, with a resolution of 2 × 2 m. In this way, final users have a dense overview of the air quality in the area, the situation within the smart zone, and distance to their homes. In the city, the heatmap resolution of the pollutant is 100 × 100 m. This justifies the second step of interpolation for providing more personalized and precise values.

In order to assess the precision, the interpolation accuracy of $PM_{10}$ has been evaluated in terms of percentage error. The error evaluation of the interpolation approach is based on the alternate exclusion of selected air quality sensor in contributing to the model and using the excluded as a true value for validation in that point on the basis of the estimation performed exploiting all the others. More precisely, for each time t we are going to estimate the error between the calculated interpolated value ${\hat{z}}_{i}\left(t\right)$. in at the position which locates the selected *i*-th sensor, with respect to the measured/real-time air quality value $z_{i}\left(t\right)$ from the *i*-th sensor. The absolute relative error of the *i*-th sensor $e_{r}^{i}$ at time t is calculated as $e_{r}^{i}\left(t\right)$ = $\frac{\left|{\hat{z}}_{i}\left(t\right)-z_{i}\left(t\right)\right|}{\left|z_{i}\left(t\right)\right|}$.

The accuracy of the whole approach is estimated by considering the same procedure for each data sensor at time t. In order to explore the performances of the interpolation approaches, two different error statistics have been calculated: The mean absolute percentage error (MAPE) per time slot and the root mean squared error (RMSE) per time slot:(4)$$\mathrm{MAPE}=\frac{1}{S}\;\overset{S}{\underset{i=1}{\sum }}e_{r}^{i}\left(t\right)$$
(5)$$\mathrm{RMSE}=\sqrt{\frac{{\left({\hat{z}}_{i}\left(t\right)-z_{i}\left(t\right)\right)}^{2}}{S}}$$
where S is the number of sensors.

About 2 months of data (from September 2019 to November 2019) have been used for the interpolation errors evaluation for the $PM_{10}$ pollutant. The error measures have been computed for:Weekends and working days, respectively: E-we, E-wd. This produces a single error by considering all devices and time slots;weekends and working days per time slots, respectively: E-we(t), E-wd(t). This produces an error value for each time slot. Working days error trend per time slots is depicted in Figure 8 and Figure 9.

The resulting error measures, MAPE and RMSE for the two methods are reported in Table 2. From the table, the Akima method resulted to be the best method in terms of validation errors.

In Figure 8 and Figure 9, RMSE (a) and MAPE (b) E-w(t) are reported for working days.

From the above figures, it is possible to see that the time bands where the error is highest are 03:00, 10:00 and 16:00, both in terms of RMSE and MAPE. A further analysis has been conducted to understand if these peaks are due to a specific device or if they depend on the time slot. For this purpose, an error $d_{i}$ has been calculated as difference between the interpolated value ${\hat{z}}_{i}\left(t\right)$ and the real value $z_{i}\left(t\right)$:(6)$$d_{i}={\hat{z}}_{i}\left(t\right)-z_{i}\left(t\right)$$
for each device i and for t= 03:00, t= 10:00 and t= 16:00.

Figure 10 and Figure 11 give an indication of how the error values $d_{i}$ are spread out, in the case of Akima method and IDW method respectively, through a box plot for each device. The distribution of each device errors data is based on a five-number summary (minimum, first quartile (Q1), median, third quartile (Q3), and maximum). The red dots represent outliers in each distribution.

## 5. Anomaly Detection of Sensor Dysfunctions

Measurement errors can be caused by a variety of factors and some countermeasures need to be taken accordingly. When a sensor error occurs, it is important to examine the cause of measurement errors thoroughly, in order to implement anomaly detection systems. This is important for ensuring stable quality in measurement and in the creation of the interpolation map. Errors in measures can be manifold:Errors caused by the measurement system: Calibration error, measurement errors originating in the measurement system, and deterioration of measurement accuracy over time (deterioration caused by wear in consumable components).Errors caused by the user: Bad positioning of the devices, mishandling of the measurement system, different degrees of skill of the users, user-specific methods of reading the scale, and turning off the device.Errors caused by environmental conditions: Deformation of the measurement target caused by rapid changes in air quality measure; measuring in locations with varying air quality measure levels.Errors caused by the measurement system or environmental conditions can be easily identified as a relevant change with respect to the average trend of the measure. When a device is left in the user’s hands, one of the most likely errors that may occur can be a dysfunction due to bad positioning or turning off the device. The detection of these types of error is important because an alert message can be sent to the user which can solve the problem.

An idea of countermeasure is to use the validation error as a detector of the device’s dysfunctions: It is possible to understand anomalies on devices comparing error trends with respect to the trend of the sensors of the same device. If the error trend is higher than the error confidence interval, it is likely to be a problem on the device. Once checked the error trend, the second step is to monitor the error on the other sensors (pollutant measure) installed on the same device. If the second measure trend error is similar to the first one, the presence of a dysfunction on the device is highly probable. This error control is quite different from a simple real-time measure trend control. A positive/negative change on the trend can be due to multiple factors and it is possible to detect it by comparing the device with the nearest one. In this case, the detected dysfunction is related to an in-correct trend over time. Such anomalies may often be useful to alert the users about a problem on the device by sending them warning messages.

To check possible dysfunctions, for each time slot t, all the points in the area of interest have an estimated/interpolated air quality value. The interpolation error and the confidence intervals are computed every 24 h based on the validation method presented in Section 4.2. The confidence interval for the average error has been computed considering a period of 2 months (working days and weekends distinctly).

### Basic Computational Approach

A computational approach for detecting dysfunction in real-time observations can be executed according to the following steps (Algorithm 2):


**Algorithm 2**
1: Input: S = sensor number
2: Input: $z_{i}\left(t\right)$ real-time value of the i-th sensor at time t
3: **for each** time t do
4:  **for**
i=1 to S
**do**
5:   **compute**
${\bar{z}}_{i}\left(t\right)$ average value of the i-th sensor at time t
6:   **compute**
$CI_{{\bar{z}}_{i}\left(t\right)}$ 95% confidence interval for the average value
7:   **compute**
${\hat{z}}_{i}\left(t\right)$ interpolated value in the i-th sensor location
8:   **compute**
$e_{r}^{i}\left(t\right)$ error interpolation measure in the i-th sensor
9:   **compute**
${\bar{e}}_{r}^{i}\left(t\right)$ average error interpolation measure
10:    **compute**
$CI_{{\bar{e}}_{r}^{i}\left(t\right)\;}\;$95% confidence interval for the average error
11:    **if**
$|z_{i}\left(t\right)-$
${\bar{z}}_{i}\left(t\right)|>CI_{z_{i}\left(t\right)}$
**then**
12:    **print** high probability of error in measure in the i-th sensor
13:    **mark** i-th sensor on the map
14:   **end if**
15:   **if**
$|z_{i}\left(t\right)-$
${\bar{z}}_{i}\left(t\right)|<CI_{z_{i}\left(t\right)}$ and $\left|{\bar{e}}_{r}^{i}\left(t\right)\right|>CI_{{\bar{e}}_{r}^{i}\left(t\right)}$
**then**
16:    **print** high probability of device dysfunction
17:    **save** the i-th sensor coordinates
18:    **send** alert message to the user
19:   **end if**
20:  **end for**
21: **end for**

Two distinct criteria are presented in the above algorithm to evaluate the possibility of a dysfunction for each sensor. Given a sensor, the first criterion considers the difference between the real-time observed/measured value $z_{i}\left(t\right)$ and the average value ${\bar{z}}_{i}\left(t\right)$ calculated for a specific time t: If the absolute value of the difference between these two measures is greater than the confidence interval for the average $CI_{{\bar{z}}_{i}\left(t\right)}$, then an anomaly may be present in the measurement. This approach allows to deduce that the type of dysfunction could derive from the context of the device position and therefore from the surrounding environment, for example, if the sensor has been positioned differently than usual (inside a building when it has always been outside), or if it could identify a possible anomaly in the measured value compared to the average of the values due to external agents (excessive level of vehicular traffic, start of some work with heavy machinery, etc.).

The second criterion investigates the case in which there are no visible differences between the measured value and the average value. In particular, it takes into account the interpolation measure ${\hat{z}}_{i}\left(t\right)$ at the position which locates the selected i-th sensor for the considered time interval t, and the absolute relative error of the i-th sensor $e_{r}^{i}$ at time t, calculated as $e_{r}^{i}\left(t\right)$ = $\frac{\left|{\hat{z}}_{i}\left(t\right)-z_{i}\left(t\right)\right|}{\left|z_{i}\left(t\right)\right|}$. If the error value is greater than the confidence interval for the average interpolation errors $CI_{{\bar{e}}_{r}^{i}\left(t\right)}$, then there is a high possibility of device failure. This means that the device is measuring something that does not comply with the surrounding devices, or a constant measurement of the sensor in the considered time interval (the device may also have been turned off). In this case, the problem may be directly investigated and corrected by the device owner once warned with messages.

Figure 12 shows $PM_{10}$ interpolation error trends in terms of absolute percentage error. Figure 12a reports an example of failure due to the second criteria presented above, in which there are no visible differences between the measured values and the average values in a 24 h interval, but the interpolation error trend for a specific device (Device 6) is higher than the confidence interval for the average interpolation errors. In Figure 12a, the trend of the device with dysfunction (Device 6) and other five devices related error trends are compared. In Figure 12b, the trends for of the five devices without any dysfunctions are compared.

## 6. Automated Heatmaps Production, Exploitation

In Snap4City solution, the heatmap production can be automatized implementing an IoT Application. The heatmaps of different kinds can be produced on the basis of any kind of geolocated data.

The automatization of heatmaps production depicted in Figure 13 reports a representation of an IoT Application in Node-RED composed of 4 different blocks. The first block named “Set R function Parameters”, is the inject node to insert the R parameters (listed below the block in the figure) in JSON (JavaScript Object Notation) format. The rounded arrow on the node shows that on that node it is possible to send the JSON created at certain frequencies chosen during configuration and modifiable at any time. The green block named “Customized Heatmap” is the Plumber-Data-analytic node to upload the R script and create a plumber instance. Note that, the Plumber-Data-analytic block return specific errors if the settings are not suitable and/or if the process for heatmap computing is already running. The orange node is a function node where it is possible to manipulate the response JSON coming from the node running the R script, since it is a JSON with the results or an error generated when creating the heatmap. The function that manipulates the JSON must be written in JavaScript and it allows to create HTML code to be displayed inside the dashboard of the Snap4city platform. The visualization can be computed using the blue node (single content node) that allows to create a widget inside a dashboard with the HTML content previously created with the function node.

To create a new customized heatmap from the source code, the principal needed/parameter are:The GPS coordinates of the area of interest (min/max latitude, min/max longitudes, etc.);The presence of a number of measured values (a set of sensors providing pollution values located in a given area, more than 5 sensors) in a specific time slot determined by a start-date and an end-date parameter in timestamp format. Each sensor has to be identified by a Service URI in the Snap4City Knowledge Base;The identification of a colormap that corresponds to the considered pollutant. If a suitable colormap is not available it is possible to create a new one by using the colormap manager positioned in the resource management tool;EPSG (Geodetic Parameter Dataset) projection that depends on the location in which the heatmap must be estimated;The value type, that is a single name or vector of possible names for the sensor’s attributes/pollutant on which compute the interpolation;The sub nature of the sensor, that is a single name that corresponds to the nature/type of the sensor of interest; andThe heatmap name and city of interest. The name of these parameters can be chosen directly by the user.

The production of the heatmap data is only the first step of the process since the heatmaps have to be transformed from a mere grid of points to GeoTIFF according to specific color map adopted for the rendering. This means that the ColorMap has to be available for the GeoTIFF creation. To this end, in Snap4City, a specific ColorMap editor and service has been created and it is exploited by an automated process that transforms the heatmap points into GeoTIFF: The so-called “heatmap GeoTIFF Generator” in Figure 2. The heatmap can be very large and at high resolution (millions or billions of areas), the distribution of the maps towards Dashboards and Mobile App is very efficient since it is performed by using a GeoServer according to WMS protocol. The produced Heatmaps can be animated if more than one instance per day is produced.

In addition, according to Figure 3, to automatize the process of the creation and use of heatmaps, there are also the following services:Alerting Assessment has been implemented by using the Snap4City Engager tool that, every time a GPS location is requested by the Mobile and Web App, is querying the “On demand fine interpolation”;Conditioned Routing has been realized by using an open source router which also queried the same “On demand fine interpolation” service via API, to get the values in specific segment of the possible routes; andGeoServer is a GIS based tool capable to provide the produced Heatmaps as GeoTIFF according to tiled exposed in the frame of the Web and Mobile Pages, via the WMS standard protocol.

## 7. Conclusions

The environmental data collected from devices hosted by city users and from data providers, have been used to provide informative view and services to citizens regarding environmental data. The paper presents an integrated solution to address different issues connected to the production of high-density real-time air quality data and services. The provided information and services can be exploited by city officials for decision making, and by final city users.

In particular, the solution allows the monitoring of an area of interest creating dense heatmaps from scattered data, computing dense grid values of pollutants and providing at the users with general information via mobile app and dashboards, also in their preferred point of interest, on paths and one demand. In this context, in the present paper, different algorithms for a scalable solution for computing pollutant related smart services are reported. The assessment of the estimation error for the dense grid of values, starting from scattered points of the sensors, gives a measure of the resulting data quality. The presented dense heatmap creation solution integrates data analytics in data flow real time processes, called IoT Applications. The solution and tools permit at the developers to create custom data analytics algorithms and see them exploited as a basis of smart solutions (e.g., for computing heatmaps, predictions). This means that the Data Analysts can focus on the algorithm while the integration with the Smart City API, the scheduling, and the management of the whole workflow are managed by the solution and can be easily addressed.

Further, this solution has the intention of putting the resulting data at disposal of the community with a double benefit: On one hand, it is possible to know the values of measured parameters on their premise; on the other hand it is possible to have a global view of data from the city with a denser sensor network, also thanks to the creation of interpolation heatmaps. However, the use of personal sensors has some disadvantages. For example, it is not possible to check if the device is correctly positioned (e.g., inside or outside the house). To solve this problem, a solution can be to monitor the trend of the interpolation mean absolute percentage errors which can help to avoid taking into account sensors’ data that can compromise the service quality. To check possible dysfunctions, one-week interpolation data has been used for the error’s evaluation, and the detected dysfunction was related to a bad trend over time. Such anomalies may often be useful to alert the user about a problem on the device. In the cases of failure detection, a warning is sent to the data ingestion process manager.

## Figures and Tables

**Figure 1 sensors-20-05435-f001:**
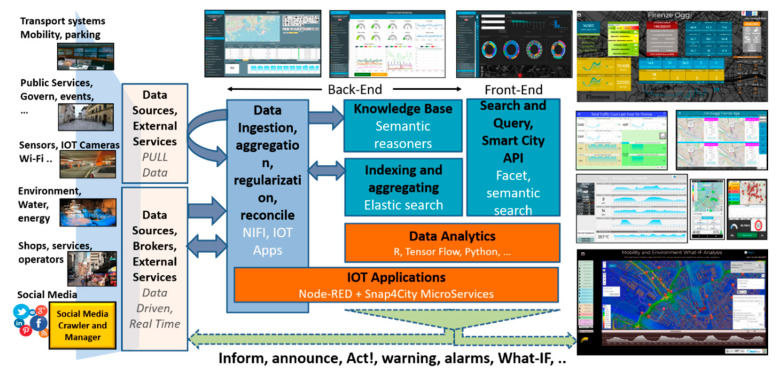
Overview of Snap4City Functional Architecture.

**Figure 2 sensors-20-05435-f002:**
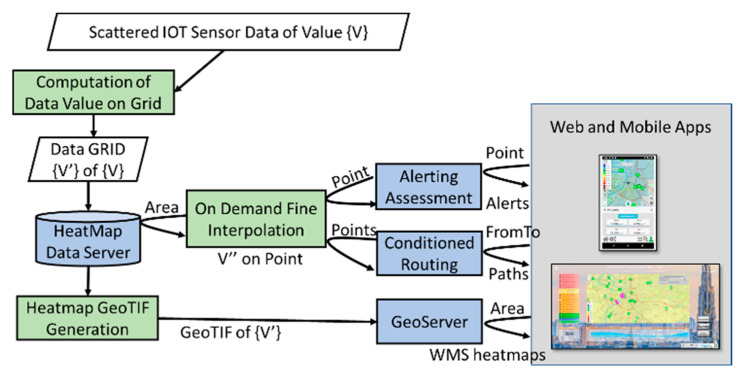
Workflow for the data computation vs. services. Green blocks are implemented as IoT Applications, cyan are tools which expose services via Smart City Application Programming Interfaces (APIs), white blocks are some of the data. Scattered IoT Sensor Data are collected via Smart City API on the Big Data Storage. Please note that all the needed services have been depicted to make the figure readable and focused.

**Figure 3 sensors-20-05435-f003:**
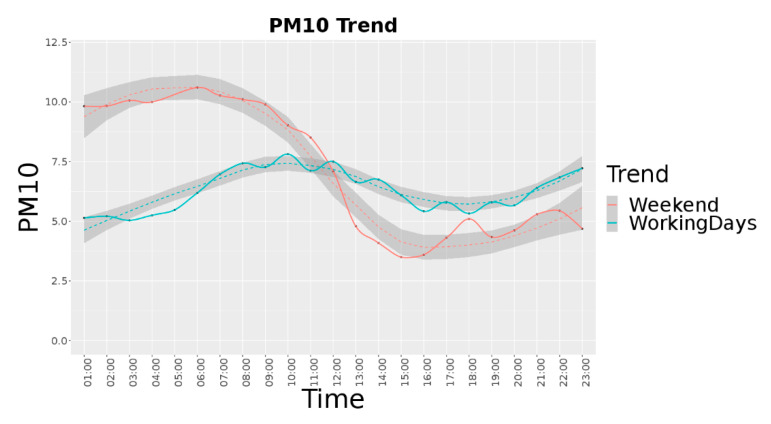
Helsinki Jätkäsaari $PM_{10}$ average trend for weekend and working days per hour.

**Figure 4 sensors-20-05435-f004:**
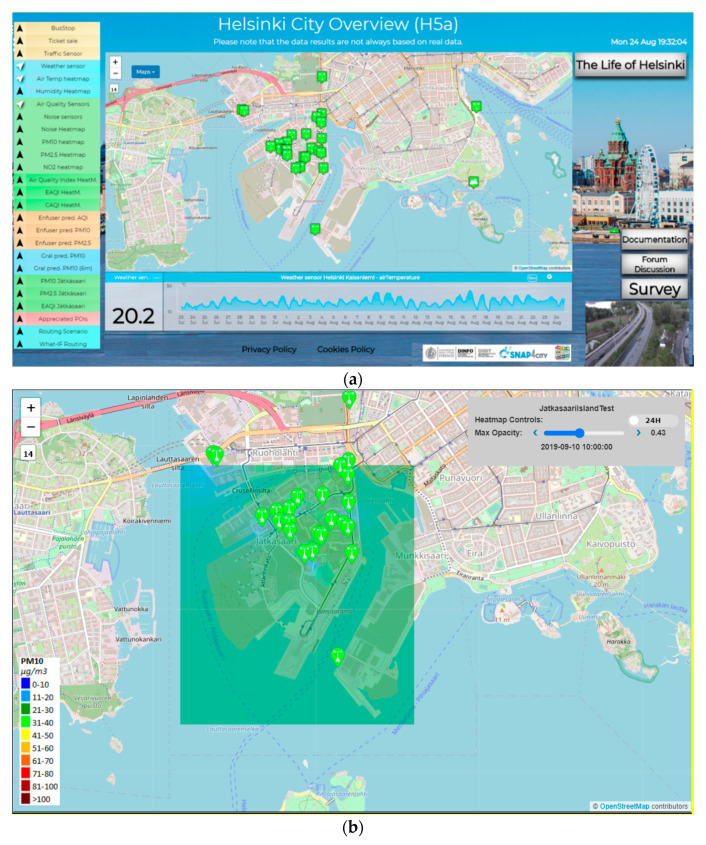
(**a**) Dashboard on Helsinki environmental aspects, the map presents an high density of sensors in the Jätkäsaari Island, (**b**) air quality $PM_{10}$ interpolation heatmap for a small area of Jätkäsaari Island (Inverse Distance Weighting (IDW) method), transparency set at 43%. The legend describes a colour map of 9 colours, while the heatmap is produced with a colour map with 150 colours.

**Figure 5 sensors-20-05435-f005:**
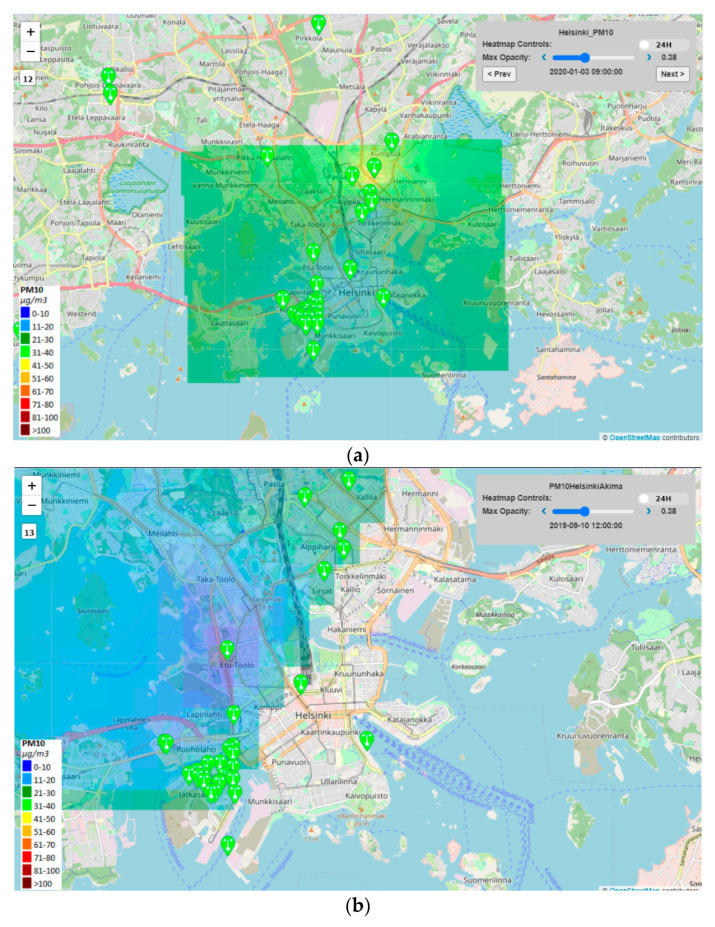
(**a**) IDW interpolation method visualization of the entire Helsinki area (regular map containing all the sensors) in the map and (**b**) Akima interpolation method visualization (irregular map—some sensors are not included in the colored area because they do not produce a measure).

**Figure 6 sensors-20-05435-f006:**
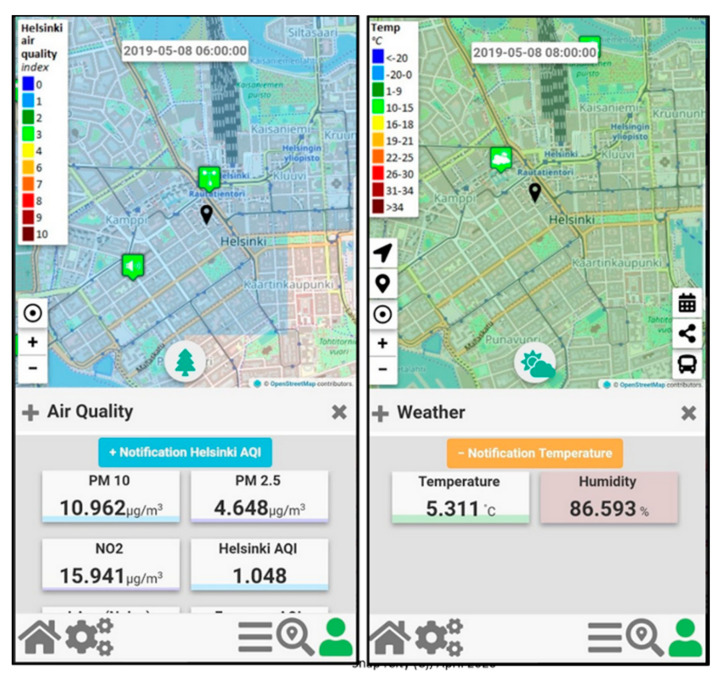
Mobile App Helsinki in a Snap, visualization of Heatmap and subscription to alerts.

**Figure 7 sensors-20-05435-f007:**
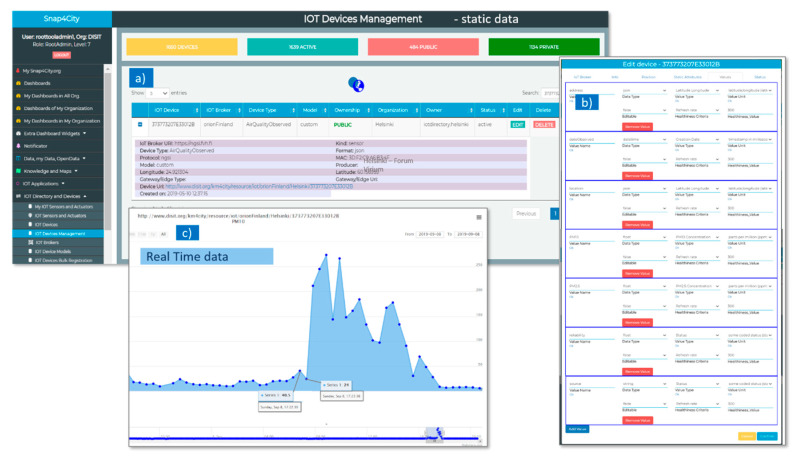
Data gathering on Snap4City Broker: (**a**,**b**) Static data related to each air quality sensor, called IoT Device in the Snap4City Platform; (**c**) History view related to the Real Time data: PM10 (and PM2.5), with frequency of 1 min.

**Figure 8 sensors-20-05435-f008:**
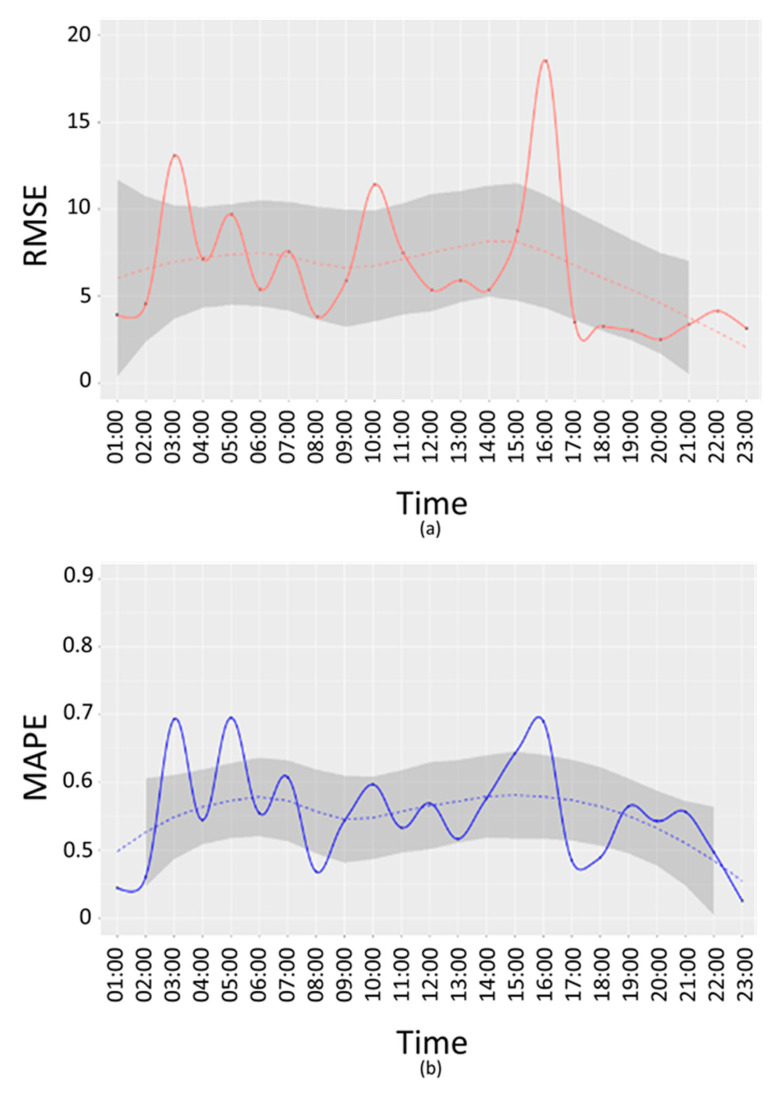
$PM_{10}$ working days root mean squared error (RMSE) (**a**) and mean absolute percentage error (MAPE) (**b**) per time slot (Akima Method).

**Figure 9 sensors-20-05435-f009:**
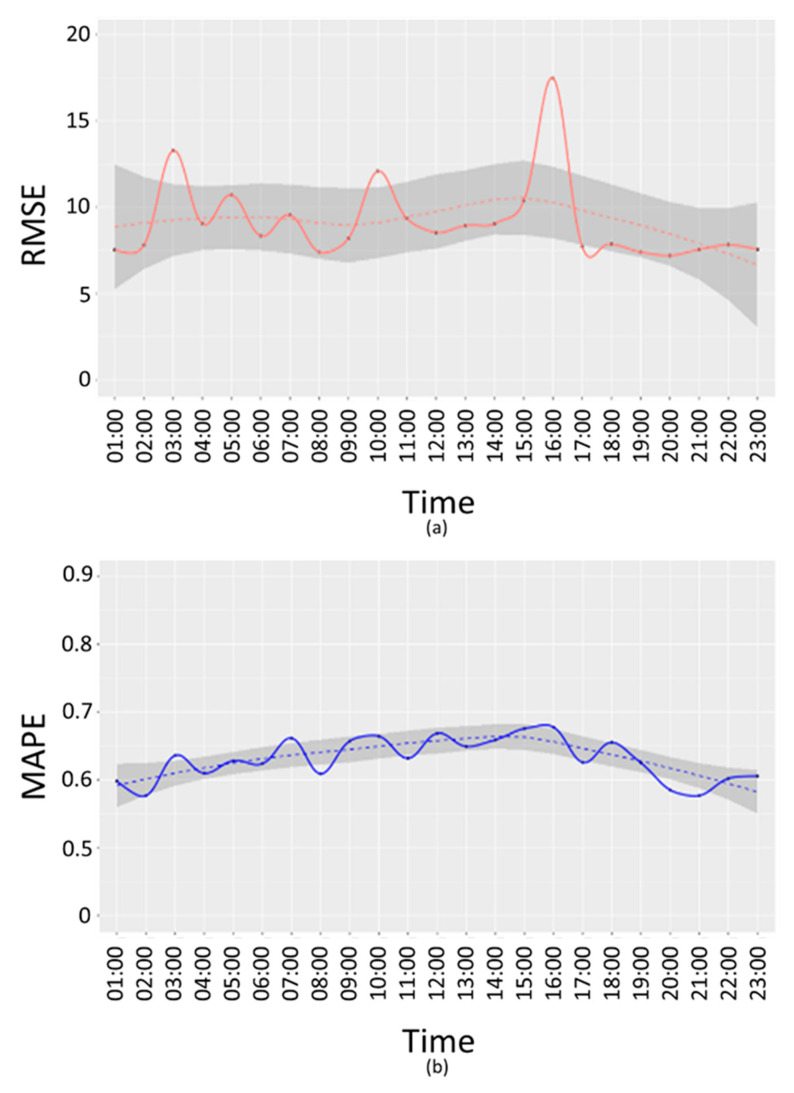
$PM_{10}$ working days RMSE (**a**) and MAPE (**b**) per time slots (IDW Method).

**Figure 10 sensors-20-05435-f010:**
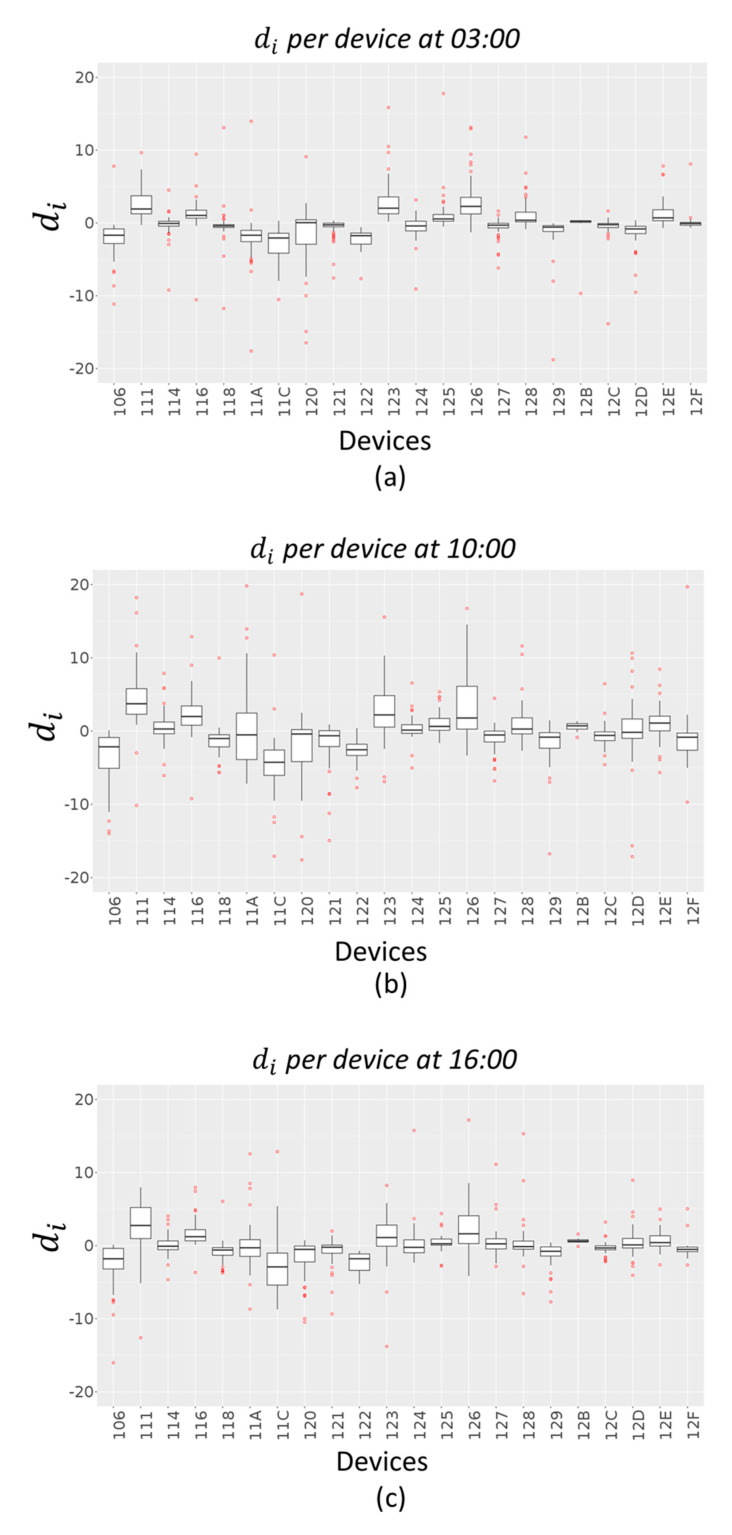
$PM_{10}$ working days error box plots per device (Akima Method) at 03:00 (**a**), 10:00 (**b**) and 16:00 (**c**).

**Figure 11 sensors-20-05435-f011:**
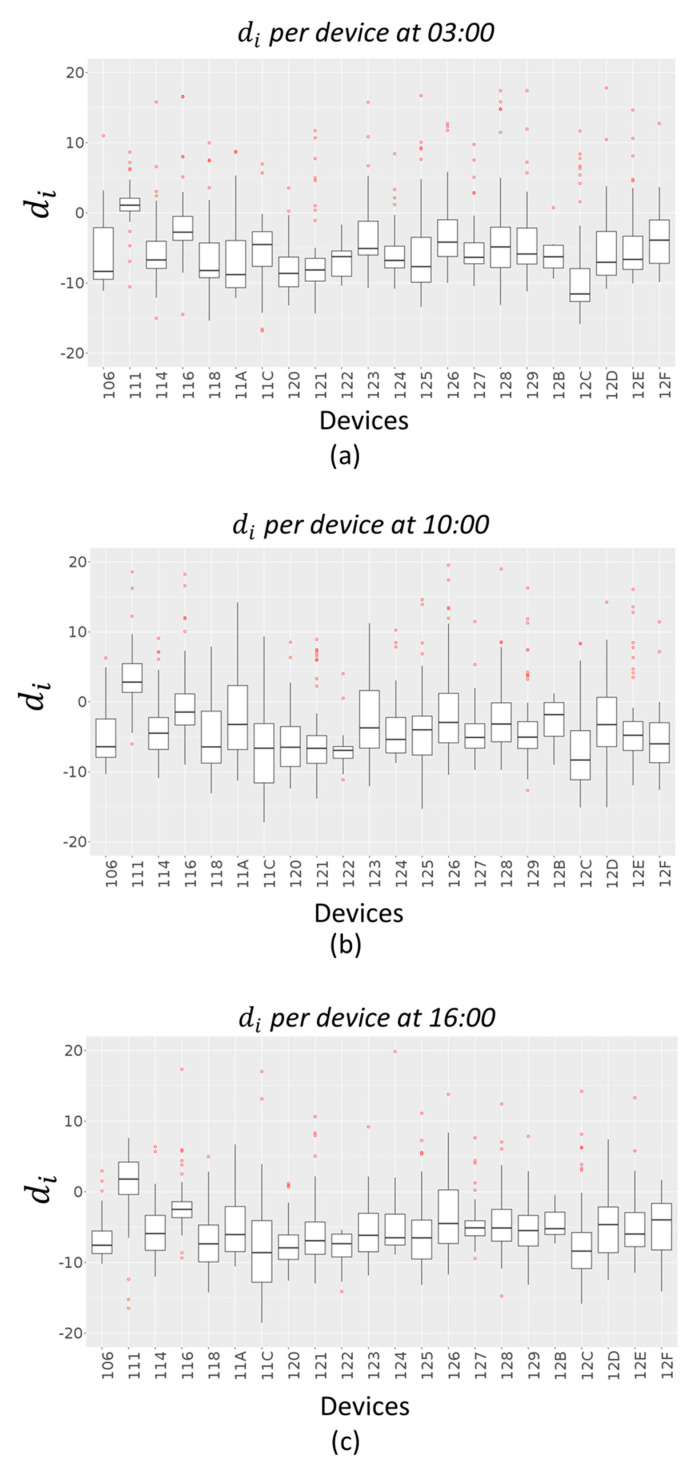
$PM_{10}$ working days error box plots per device (IDW Method) at 03:00 (**a**), 10:00 (**b**) and 16:00 (**c**).

**Figure 12 sensors-20-05435-f012:**
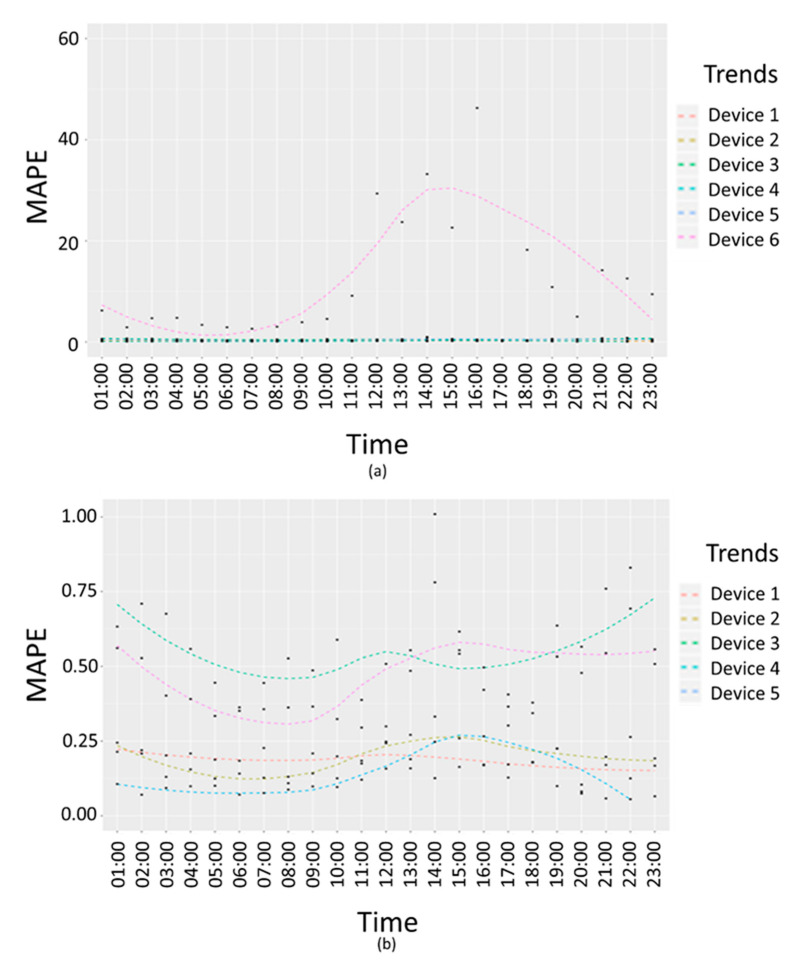
Air quality $PM_{10}$ working days interpolation error trends per hour in terms of mean absolute percentage error for (**a**) six personal devices including the device with a dysfunction; (**b**) five personal devices.

**Figure 13 sensors-20-05435-f013:**
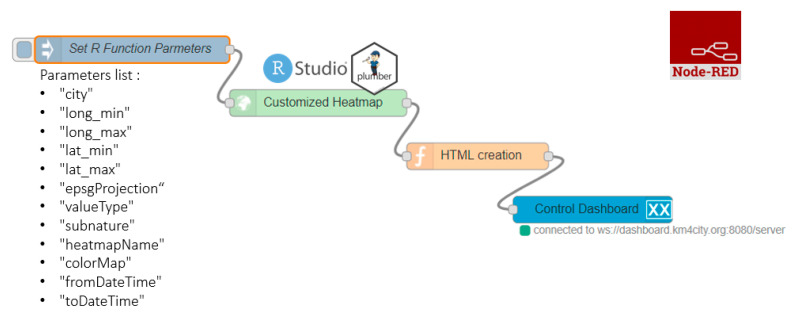
Node-RED workflow for automatization of heatmap production.

**Table 1 sensors-20-05435-t001:** Comparison among Helsinki station area and Jätkäsaari Island on Annual Means related to $PM_{10}$, $PM_{2.5}$, $NO_{x}$, and Air Quality Index/European Air Quality Index (AQI/EAQI) for a single representative sensor in each area.

Area	$\mathrm{Mean}\;\mathrm{Annual}\;PM_{10}$	$\mathrm{Mean}\;\mathrm{Annual}\;PM_{2.5}$	$\mathrm{Mean}\;\mathrm{Annual}\;NO_{x}$	Mean Annual EAQI
Helsinki station (downtown)	19.34	6.83	21.62	1.64 (Fair)
Helsinki Jätkäsaari (periphery)	13.18	4.62	15.21	1.73 (Fair)

**Table 2 sensors-20-05435-t002:** Error measures vs. interpolation methods.

Error Measures	Akima	IDW
MAPE	0.69	0.79
RMSE	8.90	12.20
MAPE-we	0.60	0.95
MAPE-wd	0.70	0.93
RMSE-we	8.60	10.70
RMSE-wd	9.70	17.00

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
