# Peer review of "High Density Real-Time Air Quality Derived Services from IoT Networks"

_sensors, 2020, doi:10.3390/s20185435_

Round 1

Reviewer 1 Report

The manuscript "Automated Production of Real-Time Air Pollution Models and Services from IOT Data Networks" submitted by Badii et al. deals with ambient air quality data interpolation to provide different services.

The topic is of interest, nevertheless the submitted manuscript is poorely organized and written. The organization of paper is confusing and reader gets lost. It is overlengthy, repeating the same information on different places throughout the manuscript, full of technical abbreviations, many of them not introduced when used for the first time. Figures in pdf file are mostly unreadable. Furthermore, for some reason, the autors included the instructions for authors in their manuscript (e.g. lines 690-725).

The manuscript is  rather  a technical report than a scientific paper.

Author Response

The topic is of interest, nevertheless the submitted manuscript is poorely organized and written. The organization of paper is confusing and reader gets lost. It is overlengthy, repeating the same information on different places throughout the manuscript, full of technical abbreviations, many of them not introduced when used for the first time.

ANSWER: The paper has been strongly focused and reorganized, shortening the first parts and improving part 4. In more details, the title has been changed to describe the focus of the paper, the abstract rewritten, the motivation and goals formalized, Figure 2 removed, section 2 reduced, section IV (part 4) fully restructured with also a new section on sensors data added, validation section extended, implementation details moved at the end of the paper, a number of figures have been added to better describe the sensors data and the  validation phases and results.

Figures in pdf file are mostly unreadable.

ANSWER: All the figures have been revised and enlarged. Please note that the size of the figures was decided by the journal editor since the paper has been gently revised by Sensor personnel, we suppose those size was suitable for publication and not for review. Sorry!

Figure 1 has been enlarged.
Figure 2 of the old version of the paper has been removed.
Figure 3 of the new version of  the paper has been added reporting the trend
Figures 4 and 5 have been enlarged
Figure 7 of the new version of the paper has been added
Figures 8 and 9 have been enlarged
Figures 10 and 11 have been added
Figure 12 of the new version of the paper (10 in the old) has been enlarged

Furthermore, for some reason, the authors included the instructions for authors in their manuscript (e.g. lines 690-725).

ANSWER: Please note that the notes have been added by the Sensor personnel, we suppose those instructions for us and not for the review. Sorry! They have been removed in  this version.

The manuscript is rather a technical report than a scientific paper.

ANSWER: The paper has been strongly focused and reorganized. In more details, the title has been changed to describe the focus of the paper, the abstract rewritten, the motivation and goals formalized, Figure 2 removed, a new section on sensor data added, validation section extended, implementation details moved at the end of the paper.

Reviewer 2 Report

The authors present a solution to the disposition of air quality sensors. The paper brings data interpolation techniques to cope with the scalability of such systems.

Required modifications:

1- The abstract is too long and a bit confusing. I suggest the authors rewrite it in a more direct approach;

2- Increase line spacing to at least 1.5pt;

3- Figures 1 to 10 have to be re-dimensioned. It is impossible to read them.

4- Tables' captions are on top, not bottom;

5- There are a lot of unfinished references in the paper with Xes all over it;

6- Appendixes A and B are missing;

7- There is an incomplete section in the paper which has to be completed by the authors;

Author Response

1- The abstract is too long and a bit confusing. I suggest the authors rewrite it in a more direct approach;

ANSWER: The paper has been strongly focused and reorganized. In more details, the title has been changed to describe the focus of the paper, the abstract rewritten, the motivation and goals formalized, Figure 2 removed, a new section on sensor data added, validation section extended, implementation details moved at the end of the paper.

2- Increase line spacing to at least 1.5pt;

ANSWER: we used the spacing is defined by the Journal Sensor style. Sorry!, probably is not the best for the review phase.

3- Figures 1 to 10 have to be re-dimensioned. It is impossible to read them.

ANSWER: All the figures have been revised and enlarged. The size of the figures was decided by the journal editor since the paper has been gently revised/reformatted by Sensor personnel, we suppose those size suitable for publication and not for review. Sorry!

Figure 1 has been enlarged.
Figure 2 of the old version of the paper has been removed.
Figure 3 of the new version of  the paper has been added reporting the trend
Figures 4 and 5 have been enlarged
Figure 7 of the new version of the paper has been added
Figures 8 and 9 have been enlarged
Figures 10 and 11 have been added
Figure 12 of the new version of the paper (10 in the old) has been enlarged

4- Tables' captions are on top, not bottom;

ANSWER: Thanks a lot, the captions of Tables have been revised according to your suggestions and journal style.

5- There are a lot of unfinished references in the paper with Xes all over it;

ANSWER: Thanks for the comment, the problems has been solved.

6- Appendixes A and B are missing;

ANSWER: Please note that the notes have been added by the Sensor personnel, we suppose those instructions for us and not for the review. Sorry! Now we removed them.

7- There is an incomplete section in the paper which has to be completed by the authors;

ANSWER: Please note that the notes have been added by the Sensor personnel, we suppose those instructions for us and not for the review. Sorry! Now we removed them.

Reviewer 3 Report

The first impression of the article is that the Authors have not thought carefully about the purpose of the article. The Authors focused on too many aspects and research areas. A large part of the text is devoted to technical issues - part 2, with a broad description of the Snap4City system, and part 3 where the authors largely describe the pseudo-code based on references. Part 4 of the article - too short and imprecise in my opinion - concerns data. In fact, the authors did not describe the essence of the obtained data. Basic information such as sampling frequency, range of data available is missing. The section on data errors lacks an extended discussion of outlier detection methods. The algorithm presented in the section entitled “Basic Computational Approach” is left without further description and comment. The results presented in the form of graphs are presented in an illegible manner. The diagram shown in Figure 1 is completely unreadable. Likewise, the following figures are prepared in a very illegible way. Besides, the descriptions of the axes in Figures 7, 8, 10 are too small. Additionally, the number of editing errors (lines 187, 188, and many others) indicates that the text was prepared without the required attention. This is also indicated by parts (from lines 694 to 725) of the text from the template provided by the Publishing House. To sum up, the Authors must define the main goal of the presented research and then focus on it in the text. The title itself - although indicating the interesting content of the text - is not reflected in the text.

Author Response

The first impression of the article is that the Authors have not thought carefully about the purpose of the article. The Authors focused on too many aspects and research areas. A large part of the text is devoted to technical issues - part 2, with a broad description of the Snap4City system, and part 3 where the authors largely describe the pseudo-code based on references. Part 4 of the article - too short and imprecise in my opinion - concerns data.

ANSWER: The paper has been strongly focused and reorganized. In more details, the title has been changed to describe the focus of the paper, the abstract rewritten, the motivation and goals formalized, Figure 2 removed, section 2 reduced, section IV (part 4) fully restructured with also a new section on sensors data added, validation section extended, implementation details moved at the end of the paper, a number of figures have been added to better describe the sensors data and the  validation phases and results.

In fact, the authors did not describe the essence of the obtained data. Basic information such as sampling frequency, range of data available is missing.

ANSWER: Thanks a lot for the suggestion. We have added for this purpose Section IV.A, and improved Section IV.B with validation.

The section on data errors lacks an extended discussion of outlier detection methods.

ANSWER: Thanks a lot for the suggestion. We have strongly improved Section IV.B on validation, presenting also the discussion on the outliers.

The algorithm presented in the section entitled “Basic Computational Approach” is left without further description and comment.

ANSWER: thanks for the comment. As regard the solution for anomaly detection, in section V.A Basic Computational Approach, we have deeply described the considered criteria to provide the evidence of the difference and dysfunctions.

The results presented in the form of graphs are presented in an illegible manner. The diagram shown in Figure 1 is completely unreadable. Likewise, the following figures are prepared in a very illegible way. Besides, the descriptions of the axes in Figures 7, 8, 10 are too small.

ANSWER: All the figures have been revised and enlarged. The size of the figures was decided by the journal editor since the paper has been gently revised/reformatted by Sensor personnel, we suppose those size was suitable for publication and not for review. Sorry the inconvenient!

Figure 1 has been enlarged.
Figure 2 of the old version of the paper has been removed.
Figure 3 of the new version of  the paper has been added reporting the trend
Figures 4 and 5 have been enlarged
Figure 7 of the new version of the paper has been added
Figures 8 and 9 have been enlarged
Figures 10 and 11 have been added
Figure 12 of the new version of the paper (10 in the old) has been enlarged

Additionally, the number of editing errors (lines 187, 188, and many others) indicates that the text was prepared without the required attention.

ANSWER: Sorry for the text. The paper has been strongly revised for structure and language.

This is also indicated by parts (from lines 694 to 725) of the text from the template provided by the Publishing House.

ANSWER: Please note that the notes have been added by the Sensor personnel, we suppose those instructions for us and not for the review. Sorry! They have been removed in  this version.

To sum up, the Authors must define the main goal of the presented research and then focus on it in the text.

The paper has been strongly focused and reorganized. In more details, the title has been changed to describe the focus of the paper, the abstract rewritten, the motivation and goals formalized, Figure 2 removed, section 2 reduced, section IV (part 4) fully restructured with also a new section on sensors data added, validation section extended, implementation details moved at the end of the paper, a number of figures have been added to better describe the sensors data and the  validation phases and results.

The title itself - although indicating the interesting content of the text - is not reflected in the text.

ANSWER: Thanks for the suggestion. The title has been changed into: “High Density Real-Time Air Quality Derived Services from IOT Networks”, while the former title was: “Automated Production of Real-Time Air Pollution Models and Services from IOT Data Networks”

Reviewer 4 Report

Line 53

In order to enrich the state of the art I suggest to insert work as:

Air quality monitoring network for tracking pollutants: The case study of Salerno city center. DOI:10.3303/CET1868012

Line 62/597

Are the sensors that send data to the system calibrated, if so what procedure is adopted to ensure good data quality?

Line 213

Increase the quality of Figure 1

Line 474

What kind of sensors are used in this specific case? What technology do they use to measure the PM10 concentration?

Line 518

Specify whether the sensors are mobile or not. This equation requires that all the sensors are placed in the same place but this is not the case, making a MAPE of different sensors is not correct, it is necessary to consider the actual different concentration of pollutants in the different points of the city.

Line 627

Were there any comparative response tests of the sensors analyzed?

Author Response

Line 53: In order to enrich the state of the art I suggest to insert work as: Air quality monitoring network for tracking pollutants: The case study of Salerno city center. DOI:10.3303/CET1868012

ANSWER: Thanks for the suggestion. It has been carefully red and cited/added.

Line 62/597 Are the sensors that send data to the system calibrated, if so what procedure is adopted to ensure good data quality?

ANSWER: Thanks a lot for the suggestion. We have added for this purpose Section IV.A in which the calibration is reported.

Line 213 Increase the quality of Figure 1

ANSWER: All the figures have been revised and enlarged. Please note that the size of the figures was decided by the journal editor since the paper has been gently revised by Sensor personnel, we suppose those size suitable for publication and not for review. Sorry!

Figure 1 has been enlarged.
Figure 2 of the old version of the paper has been removed.
Figure 3 of the new version of  the paper has been added reporting the trend
Figures 4 and 5 have been enlarged
Figure 7 of the new version of the paper has been added
Figures 8 and 9 have been enlarged
Figures 10 and 11 have been added
Figure 12 of the new version of the paper (10 in the old) has been enlarged

Line 474 What kind of sensors are used in this specific case? What technology do they use to measure the PM10 concentration?

ANSWER: Thanks a lot for the suggestion. We have added for this purpose Section IV.A in which the calibration is reported.

Line 518 Specify whether the sensors are mobile or not. This equation requires that all the sensors are placed in the same place but this is not the case, making a MAPE of different sensors is not correct, it is necessary to consider the actual different concentration of pollutants in the different points of the city.

ANSWER: ANSWER: Thanks a lot for the suggestion. We have added for this purpose Section IV.A. In details the considered sensors are not mobile. This fact has been better explained in the new version of the paper. For this reason, the use of MAPE is correct.

Line 627 Were there any comparative response tests of the sensors analyzed?

ANSWER: In the description of the basic computational approach in section V of the new version of the paper, there is also a comparative analysis of sensors dysfunction based on the obtained results in terms of MAPE (see Figure 12 of the new version of the paper).

Round 2

Reviewer 1 Report

The manuscript has been substantially improved and i believe that it might be published.

Reviewer 3 Report

Thank you very much for introducing changes to the text and accepting my proposed changes. Currently, the text seems to be coherent and clearly explains the research issue. Please note a minor editing error in Figure 7, where there is a colon instead of a period. After removing this minor error, I hope your article will be published in the publishing house.